# When the Brain Sees Beyond Pixels: Creative Brain-to-Vision Reconstruction

## Abstract

Reconstructing images from fMRI has traditionally been framed as maximizing pixel fidelity to visual input. While useful for benchmarking, this perspective overlooks what brain signals truly encode: not only perception, but also abstraction, semantics, and imagination. We introduce a frequency-informed framework for brain-to-vision generation that shifts the objective from replication to creative alignment across neural and visual domains. Our method applies graph spectral transforms to fMRI signals and masked frequency modeling to images, enabling coarse-to-fine reconstruction by selectively aligning low-, mid-, and high-frequency structures. To ground generation in meaning, we incorporate semantic priors via CLIP-text embeddings and multi-level visual features, with attention mechanisms that allow frequency-masked brain signals to interact with both reconstructions and textual cues. The model integrates pretrained VDVAE, CLIP, and diffusion backbones, while introducing three novel frequency-aligned projection layers: (i) a low-level hierarchical brain-to-vision layer, (ii) a high-level semantic brain-to-vision layer, and (iii) a brain-to-text alignment layer. The resulting generations may deviate from pixel-level ground truth yet capture emergent structures that show how the brain creatively encodes and reinterprets visual experience. By bridging frequency structures across neural, visual, and semantic modalities, our approach reframes fMRI-to-image reconstruction as a study of how humans perceive, imagine, and create, beyond simple replication.

## 1 Introduction

Decoding visual experiences from brain activity is a longstanding challenge at the intersection of neuroscience and machine learning. Functional MRI (fMRI) provides only an indirect, noisy measure of neural processes, while natural images embody rich multi-scale structure (Rakhimberdina et al., 2021). Bridging these heterogeneous representations is central not only to advancing brain-computer interfaces, but also to probing how the brain encodes perception, imagination, and abstraction. Recent progress in deep generative modeling has dramatically advanced this task (Ozcelik et al., 2022; Caselles-Dupré et al., 2024; Allen et al., 2022). By mapping neural activity into the latent space of large pretrained generators such as variational autoencoders (VAEs) or diffusion models, researchers have produced reconstructions of naturalistic faces, objects, and scenes from fMRI with unprecedented fidelity (Kim et al., 2021; Qiang et al., 2021; Zhang et al., 2021).

Latent diffusion models, in particular, have enabled highly naturalistic reconstructions by coupling coarse visual predictions with semantic refinement. Ozcelik and VanRullen (2023) introduced the Brain-Diffuser (Ozcelik & VanRullen, 2023) pipeline, in which a Very-Deep VAE (VDVAE) (Child, 2020) provides a coarse stimulus approximation, later refined by a CLIP(Radford et al., 2021)-conditioned diffusion model. Takagi and Nishimoto (2023) further demonstrated that direct mapping of fMRI signals into the latent space of a pretrained Stable Diffusion model yields reconstructions that are semantically faithful and visually sharp at $512{\times}512$ resolution (Takagi & Nishimoto, 2023), without finetuning the generator itself. While powerful, these approaches share a crucial limitation: they treat all image information uniformly, ignoring the brain's own frequency-specific organization. Neuroscience evidence (Broderick et al., 2022; Bartsch et al., 2022; Friedl & Keil, 2020) shows that visual cortex is selectively tuned to spatial frequency bands, from low-frequency global layout to high-frequency fine detail, yet current decoders (Ozcelik & VanRullen, 2023; Wang et al., 2024;

| GT | MindAligner | Brain-Diffuser | MindEye2 | MindBridge | Ours |
| --- | --- | --- | --- | --- | --- |

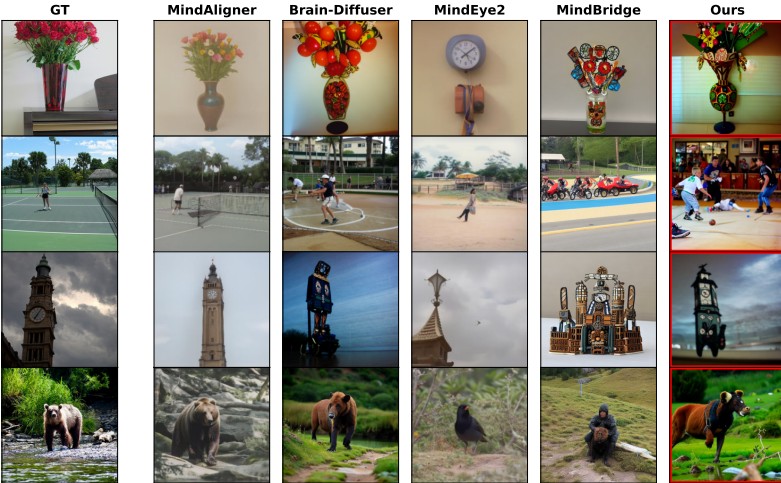

Figure 1: fMRI-to-image reconstruction with frequency-guided alignment. Comparison of four state-of-the-art methods (MindAligner, Brain-Diffuser, MindEye2, MindBridge) with our framework (Ours, red box). Prior methods often blur fine details, misrepresent object identity, or fail to capture semantic context. In contrast, our approach preserves global layout (*e.g.*, tennis court lines), captures object identity and distinctive attributes (*e.g.*, bear shape, clock tower silhouette, vase of flowers), and allows creative reinterpretation, reflecting how the brain encodes and reconstructs visual experience. By explicitly aligning neural, visual, and semantic frequency structures, our method goes beyond pixel-level replication to reveal emergent patterns in perception and imagination.

Beliy et al., 2019) collapse these heterogeneous signals into a single latent representation, diluting their interpretability and biological plausibility.

We introduce a frequency-informed framework for brain-to-vision generation that closes this gap by explicitly aligning the spectral structures of neural, visual, and semantic modalities (Palazzo et al., 2020; Van de Putte et al., 2018). On the neural side, we apply a graph spectral transform to fMRI data, embedding voxel activations into frequency components defined on the cortical graph. This decomposition yields low-, mid-, and high-frequency graph modes that compactly capture the brain's representational hierarchy. On the visual side, we adopt masked frequency modeling, dynamically filtering Fourier components (Wang et al., 2023; Li et al., 2023) of image embeddings to emphasize the scales most relevant to neural graph frequencies. By doing so, our method performs brain-to-image reconstruction in a coarse-to-fine manner, selectively aligning brain graph modes with visual spatial frequencies.

Crucially, our approach does not rely on finetuning large generative backbones. Instead, we reuse pretrained VDVAE (Child, 2020), CLIP-Vision (Radford et al., 2021), CLIP-Text (Radford et al., 2021), and diffusion modules (Xu et al., 2023), and introduce three lightweight frequency-aligned projection layers that mediate cross-modal alignment. The *low-level hierarchical brain-to-vision layer* aligns masked fMRI signals with hierarchical probabilistic features extracted by the VD-VAE encoder, capturing coarse structures and layouts. The *high-level semantic brain-to-vision layer* aligns masked fMRI signals with deterministic semantic features from the CLIP-Vision encoder, ensuring consistency with higher-order object and scene information. Finally, the *brain-to-text alignment layer* connects masked fMRI signals to CLIP-Text embeddings, allowing language priors to guide generation toward coherent and imaginative reconstructions. This design preserves the expressive power of pretrained models while introducing a biologically grounded adaptation that connects neural, visual, and semantic spaces.

Beyond replication of ground-truth stimuli, our framework reframes fMRI-to-image reconstruction as a problem of creative alignment. By conditioning on frequency-masked brain signals, enriched with textual priors, our model generates reconstructions that are both coherent and imaginative, revealing emergent structures that reflect the interpretive nature of human vision. Fig. 1 shows

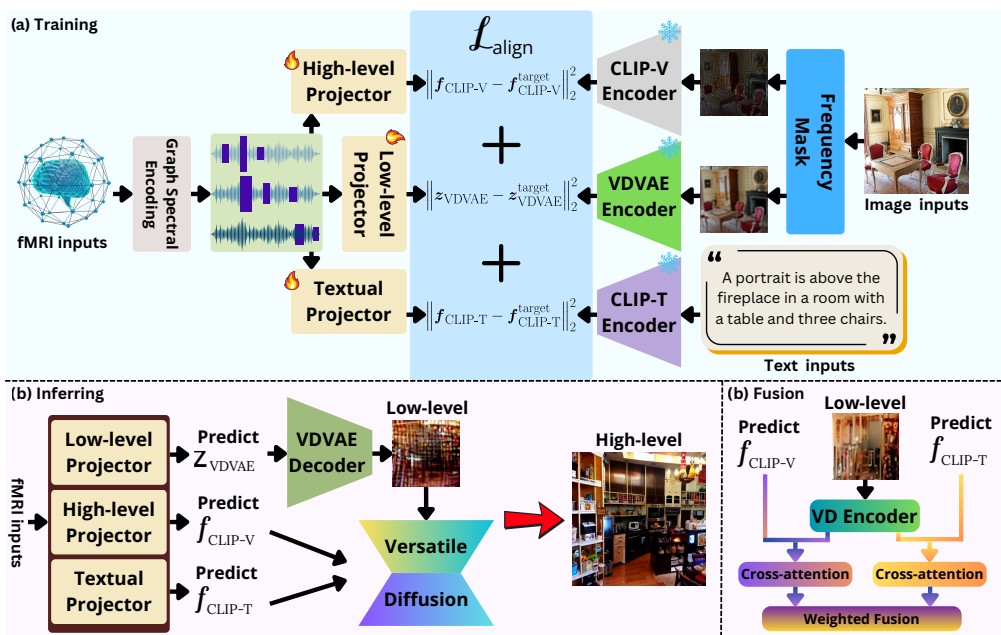

Figure 2: Overview of our frequency-informed brain-to-vision framework. (a) Training: fMRI is decomposed into graph-spectral frequency bands and mapped through three projection layers: (i) low-level projector (fMRI→VDVAE), (ii) high-level projector (fMRI→CLIP-Vision), and (iii) textual projector (fMRI→CLIP-Text), aligned with frequency-masked images and captions. (b) Fusion: the three projected features condition a pretrained diffusion model via cross-attention followed by a weighted fusion, combining structural layout (low-level), semantic content (high-level), and textual priors. (c) Inference: given new fMRI inputs, the trained projections yield reconstructions that are semantically coherent and imaginative, going beyond pixel-level replication.

our method preserves layout, captures object identity, and enables creative reinterpretation beyond pixel-level replication. In summary, our contributions are threefold:

i. **Frequency-guided neural representation.** We introduce the use of graph spectral transforms to project fMRI into frequency components, yielding a structured decomposition that parallels visual frequency representations.

ii. **Cross-modal frequency alignment.** We propose masked frequency modeling for images and demonstrate that selective alignment between brain graph modes and visual spatial frequencies improves fidelity, interpretability, and robustness of reconstructions.

iii. **Lightweight multimodal alignment layers.** We show that training only three projection layers: low-level (fMRI-VDVAE), high-level (fMRI-CLIP-Vision), and brain-to-text (fMRI-CLIP-Text), on top of frozen pretrained backbones enables reconstructions that are faithful yet creative, reframing brain decoding as exploration rather than mere replication.

Prior work on fMRI-to-image reconstruction spans diffusion-based methods (Guo et al., 2024; Ferrante et al., 2024; Chen et al., 2023; Zeng et al., 2024), cross-subject alignment (Li et al., 2024; Gong et al., 2025; Han et al., 2024; Liu et al., 2024b), and multimodal brain-conditioned generation (Xia et al., 2024; Yu et al., 2025b; Qiu et al., 2025; Yeung et al., 2025). Our contribution introduces a frequency-informed framework that explicitly bridges fMRI graph spectra with image frequency bands and semantic priors, while training only three lightweight projection layers. This distinguishes our approach from prior methods that treat all image information uniformly, providing principled interpretability and creative generation capabilities. We discuss related work in Appendix A.1 and highlight how our approach differs from existing methods.

## 2 METHOD

**Overview.** We introduce a frequency-informed framework (Fig. 2) that reconstructs images from fMRI by aligning brain activity with pretrained vision and language representations. The key idea is to operate in the frequency domain: fMRI signals are projected into the graph Fourier basis of the brain connectome (Rué-Queralt et al., 2021), yielding low-, mid-, and high-frequency components. In parallel, images are decomposed in the Fourier domain and stochastically masked, ensuring that corresponding frequency bands in brain and vision features can be explicitly aligned. Text captions provide an additional semantic prior, grounding reconstructions beyond pixel fidelity.

To establish this cross-modal alignment, we train three projection layers, each implemented as a fully connected mapping from graph-spectral fMRI features into pretrained embedding spaces: (i) low-level visual features from VDVAE, (ii) high-level semantic features from CLIP-Vision, and (iii) textual embeddings from CLIP-Text. Crucially, these layers perform forward mappings from brain activity into vision/text feature spaces, allowing fMRI signals to be expressed in the same representational domains as pretrained models without inverting their encoders.

Reconstruction proceeds in a coarse-to-fine manner. First, fMRI-aligned low-level features are decoded via VDVAE into an initial image capturing coarse structure and layout. Next, high-level semantic features and text embeddings are combined with this structural prior within the cross-attention mechanism of a pretrained Versatile Diffusion model, yielding the final reconstruction.

The framework use strong pretrained models (ImageNet(Deng et al., 2009)-pretrained VDVAE, LAION2B(Schuhmann et al., 2021)-pretrained Versatile Diffusion, and CLIP for text/vision) while introducing a novel frequency-alignment strategy that links neural, visual, and textual domains. This design provides both interpretability via frequency-specific mappings, and generative flexibility, enabling semantically coherent reconstructions that go beyond pixel-level similarity.

### 2.1 GRAPH-SPECTRAL FMRI ENCODING

We represent the brain as a graph $\mathcal{G} = (\mathcal{V}, \mathcal{E})$, where nodes $\mathcal{V}$ correspond to voxels and edges $\mathcal{E}$ encode local anatomical or functional relationships. The normalized graph Laplacian is defined as $\boldsymbol{L} = \boldsymbol{I} - \boldsymbol{D}^{-1/2}\boldsymbol{A}\boldsymbol{D}^{-1/2}$, where $\boldsymbol{A}$ is the adjacency matrix and $\boldsymbol{D}$ is the degree matrix. Its eigenvectors $\boldsymbol{U}$ ($\boldsymbol{U} \leftarrow \mathrm{eig}(\boldsymbol{L})$) form the *connectome harmonics* (Atasoy et al., 2016; 2017; Rué-Queralt et al., 2021), providing an orthonormal basis for cortical activation patterns. Given an fMRI activation vector $\boldsymbol{b} \in \mathbb{R}^{|\mathcal{V}|}$, we project it into the graph spectral domain:

$$\widehat{\boldsymbol{b}} = \boldsymbol{U}^\top \boldsymbol{b}, \quad \widehat{b}_i = \boldsymbol{U}_i^\top \boldsymbol{b}. \tag{1}$$

Eigenvectors corresponding to small eigenvalues capture smooth, low-frequency cortical patterns, while larger eigenvalues encode high-frequency, fine-grained variations. To exploit multi-scale neural information, we partition the graph spectrum into $B_1$ frequency bands: $\widehat{\boldsymbol{b}} = [\widehat{\boldsymbol{b}}_1, \widehat{\boldsymbol{b}}_2, \dots, \widehat{\boldsymbol{b}}_{B_1}]$. To improve robustness and focus on informative frequencies, we apply stochastic *frequency masking* on the spectral representation. Direct eigendecomposition is computationally expensive for large graphs; therefore, we approximate graph spectral filtering using Chebyshev polynomials (Hammond et al., 2011):

$$\boldsymbol{b}_{\text{filtered}} \approx \sum_{k=0}^{K-1} \theta_k\, T_k(\tilde{\boldsymbol{L}}), \quad \tilde{\boldsymbol{L}} = \frac{2}{\lambda_{\max}}\boldsymbol{L} - \boldsymbol{I}, \tag{2}$$

where $K$ is the polynomial order, $k = 0, \dots, K-1$ indexes the Chebyshev terms, $T_k(\cdot)$ are Chebyshev polynomials, and $\theta_k$ are the coefficients for each term. In principle, $\theta_k$ are learnable parameters that can be optimized via gradient descent to emphasize specific graph frequencies. In our current implementation, we initialize them as uniform values and apply stochastic masking within chosen bands, providing a computationally efficient yet flexible approximation:

$$\theta_k \leftarrow 0, \quad \forall k \in \mathcal{M}_f, \tag{3}$$

where $\mathcal{M}_f$ denotes the set of Chebyshev indices corresponding to the masked frequency band $f \in \{\text{low, mid, high, even}\}$. Masking can target low-, mid-, or high-frequency bands, zeroing a fraction of coefficients within the chosen band while leaving others intact. Alternatively, *even* masking randomly zeros coefficients uniformly across all frequencies, without privileging any specific band.

This design enables controlled exploration of how distinct spectral components contribute to brain-to-vision reconstruction.

The largest eigenvalue $\lambda_{\max}$ is estimated via power iteration and used to scale the Laplacian spectrum (Mohar et al., 1991) to $[-1, 1]$, ensuring numerical stability for the Chebyshev recursion. Mask ratio $\alpha_1$, number of bands $B_1$, and band type $f$ are tunable hyperparameters that allow systematic exploration of frequency contributions.

This approach offers four key advantages: (i) Separating low-, mid-, and high-frequency components mirrors the brain's hierarchy, where early visual areas prefer intermediate frequencies and higher areas capture global, low-frequency structure. (ii) Chebyshev approximation enables efficient filtering of both raw and normalized fMRI. Mask ratios and band partitions are tunable, revealing which cortical scales drive reconstruction. (iii) Frequency-masked fMRI can be directly matched to image frequency bands, supporting principled low-, mid-, and high-frequency correspondence and enabling reconstructions that are both semantically coherent and imaginative. (iv) By avoiding full eigendecomposition, the method scales to tens of thousands of voxels while retaining the ability to explore multi-band interactions, making it practical for large fMRI datasets.

## 2.2 Image Frequency Masking

To enable cross-modal alignment with fMRI signals, we decompose each image $\boldsymbol{I} \in \mathbb{R}^{H \times W \times C}$ into its 2D Fourier components:

$$\boldsymbol{F}\{\boldsymbol{I}\}(u, v) = \sum_{x=0}^{H-1} \sum_{y=0}^{W-1} \boldsymbol{I}(x, y) \, e^{-2\pi i (ux/H + vy/W)}, \qquad (4)$$

where $(x, y)$ are spatial pixel coordinates, $(u, v)$ index spatial frequencies, and $H, W$ are the image height and width. The resulting spectrum $\boldsymbol{F}\{\boldsymbol{I}\} \in \mathbb{C}^{H \times W}$ has the same resolution as the input image. We partition the frequency spectrum into $B_2$ bands, grouped into low-, mid-, and high-frequency ranges. For each band $f$, we construct a binary mask $\boldsymbol{M}_f \in \{0, 1\}^{H \times W}$ in the frequency domain that isolates the desired frequency range. Frequency-filtered reconstructions are then obtained as

$$\boldsymbol{I}_f = \boldsymbol{F}^{-1}\big(\boldsymbol{M}_f \odot \boldsymbol{F}\{\boldsymbol{I}\}\big), \quad f \in \{\text{low}, \text{mid}, \text{high}, \text{even}\}, \qquad (5)$$

where $\odot$ denotes element-wise multiplication and $\boldsymbol{F}^{-1}$ is the inverse Fourier transform.

During training, stochastic frequency masking is applied to enforce robustness and encourage multi-scale integration. Masking strategies are defined as follows: low masks primarily low-frequency bands, mid targets intermediate bands, high masks high-frequency bands, and even randomly masks coefficients uniformly across all frequency bands without privileging any range. The *mask ratio* $\alpha_2 \in [0, 1]$ specifies the fraction of coefficients set to zero within the chosen strategy. These hyperparameters, along with the number of bands $B_2$, are tunable for systematic exploration.

This design encourages the network to learn hierarchical visual representations: low frequencies encode coarse shape and global layout, mid frequencies capture edges and patterns, and high frequencies represent fine textures. Practically, frequency masking serves as both a regularizer (preventing overfitting to dominant bands) and as a cross-modal alignment mechanism, directly matching image frequencies with fMRI graph-spectral bands.

## 2.3 Frequency-aligned Projection

A core component of our framework is the set of three frequency-aligned projection layers, which map graph-spectral fMRI features into pretrained vision and language embedding spaces.

**Low-level visual projection.** The first projection layer maps low-frequency fMRI components to the latent space of a pretrained VDVAE. Formally, let $\boldsymbol{b}_{\text{low}} \in \mathbb{R}^{|\mathcal{V}|}$ denote the low-frequency graph-spectral fMRI vector. The low-level projection layer $\Phi_{\text{VDVAE}} : \mathbb{R}^{|\mathcal{V}|} \to \mathbb{R}^{d_{\text{VDVAE}}}$ is implemented as a fully connected layer:

$$\boldsymbol{z}_{\text{VDVAE}} = \Phi_{\text{VDVAE}}(\boldsymbol{b}_{\text{low}}) = \boldsymbol{W}_{\text{low}} \boldsymbol{b}_{\text{low}} + \boldsymbol{b}_{\text{low}}^{\text{bias}}, \qquad (6)$$

---

**Algorithm 1** Training frequency-aligned projection layers

---

**Require:** Dataset $\mathcal{D} = \{(\boldsymbol{b}, \boldsymbol{I}, \text{caption})\}$, pretrained models $(\mathcal{D}_{\text{VDVAE}}, \text{CLIP-V}, \text{CLIP-T})$, projection layers $(\Phi_{\text{VDVAE}}, \Phi_{\text{CLIP-V}}, \Phi_{\text{CLIP-T}})$

1: **for** each batch $(\boldsymbol{b}, \boldsymbol{I}, \text{caption}) \in \mathcal{D}$ **do**

2:     **Graph-spectral fMRI encoding and masking:**

$$\boldsymbol{L} = \boldsymbol{I} - \boldsymbol{D}^{-1/2}\boldsymbol{A}\boldsymbol{D}^{-1/2}, \quad \boldsymbol{U} \leftarrow \text{eig}(\boldsymbol{L})$$

$$\widehat{\boldsymbol{b}} = \boldsymbol{U}^{\top}\boldsymbol{b}, \quad \{\widehat{\boldsymbol{b}}_f\}_{f \in \{\text{low, mid, high}\}} \text{ partitioned from } \widehat{\boldsymbol{b}}$$

$$\boldsymbol{b}_f \leftarrow \text{ChebyshevApprox}(\widehat{\boldsymbol{b}}_f, \{\theta_k\}, \mathcal{M}_f)$$

3:     **Image frequency masking:**

$$\boldsymbol{I}_f = \boldsymbol{F}^{-1}(\boldsymbol{M}_f \odot \boldsymbol{F}\{\boldsymbol{I}\}) \quad f \in \{\text{low, mid, high, even}\}$$

4:     **Extract target embeddings from pretrained models:**

$$\boldsymbol{z}_{\text{VDVAE}}^{\text{target}} \leftarrow \mathcal{D}_{\text{VDVAE}}(\boldsymbol{I}_{\text{low}}), \quad \boldsymbol{f}_{\text{CLIP-V}}^{\text{target}} \leftarrow \text{CLIP-V}(\boldsymbol{I}_{\text{high}}), \quad \boldsymbol{f}_{\text{CLIP-T}}^{\text{target}} \leftarrow \text{CLIP-T}(\text{caption})$$

5:     **Compute predicted embeddings via projection layers:**

$$\boldsymbol{z}_{\text{VDVAE}} \leftarrow \Phi_{\text{VDVAE}}(\boldsymbol{b}_{\text{low}}), \quad \boldsymbol{f}_{\text{CLIP-V}} \leftarrow \Phi_{\text{CLIP-V}}(\boldsymbol{b}_{\text{high}}), \quad \boldsymbol{f}_{\text{CLIP-T}} \leftarrow \Phi_{\text{CLIP-T}}(\boldsymbol{b})$$

6:     **Compute frequency-alignment loss:**

$$\mathcal{L}_{\text{align}} = \|\boldsymbol{z}_{\text{VDVAE}} - \boldsymbol{z}_{\text{VDVAE}}^{\text{target}}\|_2^2 + \|\boldsymbol{f}_{\text{CLIP-V}} - \boldsymbol{f}_{\text{CLIP-V}}^{\text{target}}\|_2^2 + \|\boldsymbol{f}_{\text{CLIP-T}} - \boldsymbol{f}_{\text{CLIP-T}}^{\text{target}}\|_2^2$$

7:     **Update trainable projection layers:**

$$\min_{\Phi_{\text{VDVAE}}, \Phi_{\text{CLIP-V}}, \Phi_{\text{CLIP-T}}} \mathcal{L}_{\text{align}}$$

8: **end for**

---

where $d_{\text{VDVAE}}$ is the dimension of flattened VDVAE latents. The predicted latents $\boldsymbol{z}_{\text{VDVAE}}$ are decoded via the pretrained VDVAE decoder to produce an initial coarse image that captures structural layout and low-level visual patterns.

**High-level semantic visual projection.** The second projection layer aligns mid- and high-frequency fMRI components $\boldsymbol{b}_{\text{high}} \in \mathbb{R}^{|\mathcal{V}|}$ with the feature space of a pretrained CLIP-Vision encoder. This layer, $\Phi_{\text{CLIP-V}} : \mathbb{R}^{|\mathcal{V}|} \to \mathbb{R}^{d_{\text{CLIP-V}}}$, is also implemented as a fully connected layer:

$$\boldsymbol{f}_{\text{CLIP-V}} = \Phi_{\text{CLIP-V}}(\boldsymbol{b}_{\text{high}}) = \boldsymbol{W}_{\text{high}}\boldsymbol{b}_{\text{high}} + \boldsymbol{b}_{\text{high}}^{\text{bias}}. \tag{7}$$

The predicted visual embeddings $\boldsymbol{f}_{\text{CLIP-V}}$ provide high-level semantic information, such as object identity and scene context, to guide the generative process.

**Textual semantic projection.** The third layer maps the full graph-spectral fMRI vector $\boldsymbol{b} \in \mathbb{R}^{|\mathcal{V}|}$ to the embedding space of a pretrained CLIP-Text encoder. Denoting this projection as $\Phi_{\text{CLIP-T}} : \mathbb{R}^{|\mathcal{V}|} \to \mathbb{R}^{d_{\text{CLIP-T}}}$:

$$\boldsymbol{f}_{\text{CLIP-T}} = \Phi_{\text{CLIP-T}}(\boldsymbol{b}) = \boldsymbol{W}_{\text{text}}\boldsymbol{b} + \boldsymbol{b}_{\text{text}}^{\text{bias}}, \tag{8}$$

these embeddings act as semantic priors, guiding the diffusion model to generate images consistent with conceptual and linguistic content.

We train the three frequency-aligned projection layers using a batch-wise procedure that maps graph-spectral fMRI features to pretrained vision and text embeddings while enforcing cross-modal frequency alignment (see Algorithm 1).

## 2.4 CROSS-MODAL FUSION VIA VERSATILE DIFFUSION

Once the frequency-aligned projection layers produce their respective embeddings, reconstruction is performed via a pretrained Versatile Diffusion (VD) model, which fuses low-level visual, high-level semantic, and textual information through cross-attention.

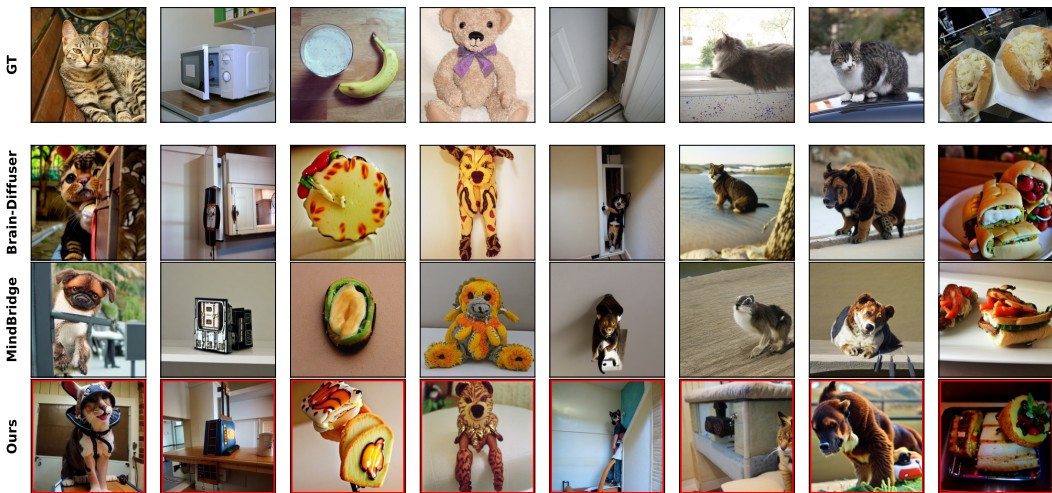

Figure 3: Qualitative comparison of fMRI reconstructions. Our frequency-informed method preserves global layout and fine semantic details better than Brain-Diffuser and MindBridge.

The reconstruction proceeds in a coarse-to-fine manner. The predicted VDVAE latents $z_{\text{VDVAE}}$ are decoded via the pretrained VDVAE decoder $\mathcal{D}_{\text{VDVAE}}$ to produce a coarse initial image:

$$\hat{I}_{\text{low}} = \mathcal{D}_{\text{VDVAE}}(z_{\text{VDVAE}}). \tag{9}$$

This image captures global structure and low-frequency visual information corresponding to coarse brain patterns. The coarse image $\hat{I}_{\text{low}}$ is encoded by the VD encoder $\mathcal{E}_{\text{VD}}$ to obtain low-level visual conditioning features:

$$u_{\text{im}} = \mathcal{E}_{\text{VD}}(\hat{I}_{\text{low}}), \tag{10}$$

which will be used as conditioning in the diffusion denoising process. In parallel, the high-level semantic and textual embeddings, $f_{\text{CLIP-V}}$ and $f_{\text{CLIP-T}}$, provide cross-modal conditioning:

$$c_{\text{im}} = f_{\text{CLIP-V}}, \quad c_{\text{tx}} = f_{\text{CLIP-T}}. \tag{11}$$

During each denoising step $t$, the VD U-Net $\mathcal{U}_t$ integrates low-level image features and semantic embeddings through cross-attention:

$$\hat{I}_t = \mathcal{U}_t\big(x_t \mid u_{\text{im}}, c_{\text{im}}, c_{\text{tx}}; \lambda_{\text{mix}}\big), \tag{12}$$

where $x_t$ is the noisy image at step $t$, and $\lambda_{\text{mix}} \in [0, 1]$ controls the relative contribution of visual versus textual conditioning. The embeddings $u_{\text{im}}, c_{\text{im}}, c_{\text{tx}}$ enter the U-Net via its frozen cross-attention modules, enabling frequency- and semantics-aware reconstruction. After $T$ denoising steps, the final reconstructed image is

$$\hat{I} = \hat{I}_T, \tag{13}$$

which integrates structural, semantic, and textual information. Preserving frequency-specific alignment ensures that each cortical scale contributes to corresponding visual and textual features, producing interpretable and high-fidelity reconstructions.

Unlike prior approaches (Ozcelik & VanRullen, 2023; Takagi & Nishimoto, 2023) that train per-slot linear regressors, our method implements fully differentiable, end-to-end projection layers. The fusion is performed inside a frozen pretrained diffusion model, preserving the generative prior while allowing explicit control over frequency-aligned brain-to-vision mappings. The use of frequency-specific embeddings ensures that each cortical scale contributes meaningfully to different visual and semantic aspects of reconstructed image, providing both interpretability and reconstruction fidelity.

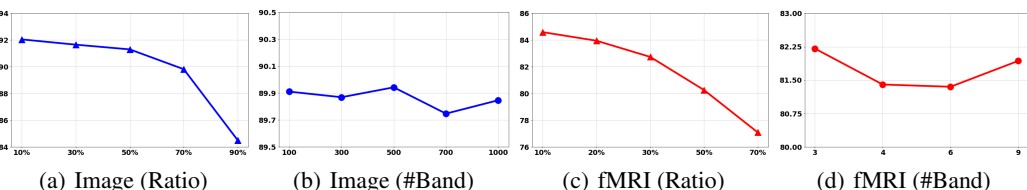

Figure 4: Hyperparameter evaluation of masking ratio and number of frequency bands for input images and fMRI. Vertical axes indicate CLIP score.

## 3 EXPERIMENT

### 3.1 SETUP

We conducted experiments using the *Natural Scenes Dataset* (NSD) (Allen et al., 2022), which provides high-resolution 7T fMRI data from subjects viewing thousands of natural images. Following prior work, we selected four participants (subj01, subj02, subj05, subj07) who completed the full protocol and used voxel-wise beta estimates from standard GLM preprocessing with denoising and regularization. Trial-averaging was applied to match previous studies (Ozcelik & VanRullen, 2023).

During training, fMRI inputs were stochastically masked with higher weights on mid-frequency components (20% masking, 4 spectral bands), while image inputs emphasized high-frequency components (10% masking, 500 spectral bands). Experiments ran on two NVIDIA Tesla V100 GPUs (32GB each) and required roughly eight hours. Pretrained generative backbones remained frozen, with only the projection layers from brain activity to latent feature spaces optimized, ensuring controlled, reproducible evaluation and computational efficiency.

### 3.2 EVALUATION

**Hyperparameter evaluation.** Fig. 4 presents the effect of masking ratio and number of frequency bands on reconstruction quality. For images, increasing the masking ratio consistently reduces the CLIP score, and a similar trend is observed for fMRI. We find the optimal number of bands to be 500 for images, while for fMRI, performance peaks at bands 3 and 9. This disparity highlights the greater difficulty of modeling fMRI signals compared to images.

**Qualitative comparison of reconstructions.** Figure 3 compares our method against Brain-Diffuser (Ozcelik & VanRullen, 2023) and MindBridge (Wang et al., 2024) across diverse stimuli. While baseline methods capture either coarse structure (Brain-Diffuser) or semantic plausibility (Mind-Bridge), they often fail to preserve both simultaneously. Brain-Diffuser tends to produce visually coherent but semantically ambiguous generations (*e.g.*, distorted fruit and plush toys), whereas MindBridge frequently yields semantically biased reconstructions (*e.g.*, generic dogs for cats) that neglect global layout.

In contrast, our frequency-informed framework achieves more faithful and interpretable reconstructions. By explicitly aligning fMRI graph-spectral components with visual frequency bands, our model preserves coarse scene layout (*e.g.*, spatial arrangement of kitchen appliances, cat positions on windowsills) while also capturing fine semantic details (*e.g.*, feline identity, stuffed toy texture, sandwich ingredients). Integration of semantic priors via CLIP-Text and CLIP-Vision further grounds generation, avoiding mode collapse toward overly generic categories. Notably, our reconstructions show creative reinterpretations that remain consistent with neural input, demonstrating how frequency-guided alignment enables reconstructions that are not only perceptually accurate but also imaginative, reflecting the interpretive nature of human vision.

**Insights from image frequency masking.** Table 1 demonstrates that frequency-specific masking strongly influences fMRI-to-image reconstruction. High-frequency inputs yield the best low-level and semantic fidelity, capturing fine details and object-specific attributes, while low- and mid-frequency inputs primarily encode coarse layout and global scene structure. Mismatched inputs (Low-High, High-Low) highlight the complementary roles of different frequencies: combining low-frequency structural information with high-frequency semantic features balances layout preservation

| Image masked | | Low-Level | | | | High-Level | | | |
|---|---|---|---|---|---|---|---|---|---|
| VDVAE | CLIP-V | PixCorr ↑ | SSIM ↑ | AlexNet(2) ↑ | AlexNet(5) ↑ | Inception ↑ | CLIP ↑ | EffNet-B ↓ | SwAV ↓ |
| Low | Low | 0.250 | 0.289 | 92.1% | 95.0% | 85.1% | 88.2% | 0.828 | 0.488 |
| Mid | Mid | 0.298 | 0.346 | 95.5% | 96.9% | 88.0% | 91.3% | 0.792 | 0.439 |
| High | High | 0.309 | **0.357** | **96.4%** | **97.2%** | **88.3%** | **92.2%** | **0.771** | **0.421** |
| Even | Even | 0.288 | 0.332 | 94.8% | 96.5% | 87.0% | 90.7% | 0.803 | 0.453 |
| Low | High | 0.273 | 0.323 | 94.8% | 96.7% | **88.3%** | 92.0% | 0.777 | 0.426 |
| High | Low | **0.313** | 0.335 | 94.1% | 96.1% | 85.8% | 88.5% | 0.818 | 0.476 |

Table 1: Ablation study on image frequency masking for VDVAE and CLIP-Vision inputs. Six masking strategies are evaluated using low- and high-level metrics. Results show that frequency-specific masking affects both structural fidelity and semantic alignment, highlighting the distinct contributions of spatial frequency bands to reconstruction quality.

| Brain-to-Vision | | Low-Level | | | | High-Level | | | |
|---|---|---|---|---|---|---|---|---|---|
| fMRI | VDVAE-CLIP-V | PixCorr ↑ | SSIM ↑ | AlexNet(2) ↑ | AlexNet(5) ↑ | Inception ↑ | CLIP ↑ | EffNet-B ↓ | SwAV ↓ |
| Low | High-High | **0.201** | **0.335** | **88.0%** | 90.7% | 79.6% | 85.4% | **0.851** | **0.488** |
| Low | Low-High | 0.167 | 0.291 | 84.3% | 89.1% | 79.2% | 85.4% | 0.859 | 0.498 |
| Mid | High-High | **0.201** | 0.334 | 87.3% | **90.9%** | 80.0% | **85.7%** | 0.853 | **0.488** |
| Mid | Low-High | 0.165 | 0.291 | 83.5% | 88.7% | 80.0% | 84.8% | 0.861 | 0.497 |
| High | High-High | 0.196 | 0.333 | 86.8% | 90.6% | 79.2% | 85.1% | 0.859 | 0.490 |
| High | Low-High | 0.162 | 0.288 | 83.5% | 88.4% | 78.8% | 84.4% | 0.864 | 0.502 |
| Even | High-High | 0.190 | 0.333 | 85.8% | 89.3% | 78.6% | 84.9% | 0.860 | 0.496 |
| Even | Low-High | 0.157 | 0.289 | 81.9% | 87.4% | 78.5% | 84.3% | 0.866 | 0.505 |

Table 2: Ablation study on graph-spectral fMRI encoding and brain-to-vision alignment. Different fMRI frequency bands (Low, Mid, High, Even) are tested with VDVAE-CLIP-V input configurations (High-High, Low-High). Results show that aligning neural and visual frequencies improves both structural and semantic reconstruction, showing how distinct cortical bands contribute to perceptual and conceptual aspects of visual experience.

and object identity. Even masking performs moderately, underscoring the importance of selective frequency alignment. These results provide evidence that our frequency-guided framework effectively leverages cortical graph-spectral signals to reconstruct both structural and semantic aspects of stimuli, producing images that preserve global organization, capture meaningful object features, and allow creative reinterpretation, revealing how the brain encodes and reconstructs visual experience beyond pixel-level replication.

**Impact of neural frequency alignment.** Table 2 shows the impact of graph-spectral fMRI encoding on brain-to-vision reconstruction. High-frequency fMRI components generally improve both low-level structural metrics and high-level semantic metrics, while low- and mid-frequency bands contribute more to coarse layout and scene organization. Comparing VDVAE-CLIP-V input configurations, High-High consistently outperforms Low-High, indicating that aligning brain and visual frequency bands enhances reconstruction fidelity. These results highlight the complementary roles of neural frequency bands: low frequencies support global structure, high frequencies capture fine details and semantic content, and their alignment enables images that reflect both perceptual accuracy and creative reinterpretation of visual experience.

## 4 CONCLUSION

We introduced a frequency-informed framework for fMRI-to-image reconstruction that aligns neural, visual, and semantic representations across low-, mid-, and high-frequency components. By combining graph-spectral fMRI encoding, masked image frequency modeling, and lightweight projections into pretrained VDVAE, CLIP, and diffusion models, our approach achieves coarse-to-fine reconstructions that preserve global layout, capture object details, and enable creative reinterpretation. Ablation studies show that different neural frequencies contribute complementary information, and aligning cortical and visual frequencies enhances both structural and semantic fidelity. Overall, our work reframes fMRI decoding as a study of how the brain perceives and imagines, producing interpretable, semantically rich, and creatively informed reconstructions.

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

# A  APPENDIX

## A.1  RELATED WORK

**Diffusion-based fMRI-to-image reconstruction.** Deep generative models have driven recent advances in reconstructing naturalistic images from fMRI. Brain-Diffuser (Ozcelik & VanRullen, 2023) introduces a two-stage pipeline: a VDVAE produces a coarse visual layout from fMRI, and a Versatile Diffusion model conditioned on CLIP features refines semantic details. Similarly, Mind-Eye (Scotti et al., 2023) maps fMRI activity into CLIP image embeddings and leverages diffusion priors, while MindEye2 (Scotti et al., 2024) pretrains across subjects and fine-tunes a Stable Diffusion XL unCLIP decoder for efficient subject-specific adaptation. NeuroPictor (Huo et al., 2024) further modulates diffusion using fMRI, combining shared multi-subject pretraining with semantic and structural conditioning. Our work differs in three key aspects. First, instead of uniformly processing neural signals, we decompose fMRI into frequency-specific graph modes and align them with corresponding image frequency bands using masked frequency modeling. Second, we incorporate pretrained VDVAE and CLIP encoders without finetuning, training only three lightweight frequency-aligned projection layers: a low-level hierarchical brain-to-vision layer for coarse visual structures, a high-level semantic brain-to-vision layer for abstract features, and a brain-to-text alignment layer for semantic guidance. Third, this design enables coarse-to-fine reconstruction while maintaining interpretability of which frequency components drive the generated image.

**Cross-subject brain decoding.** Aligning fMRI representations across participants is a major challenge. MindBridge (Wang et al., 2024) uses adaptive max-pooling and cyclic reconstruction loss for subject-invariant embeddings, while MindAligner (Dai et al., 2025) learns an explicit Brain Transfer Matrix with multi-level functional alignment to project fMRI from any subject into a reference space. Xu *et al.* (Xu et al., 2025) propose a bidirectional autoencoder with subject-bias modulation and semantic refinement for ControlNet+Stable Diffusion generation. MindCustomer (Yu et al., 2025a) integrates brain signals with external visual context using an image-brain translator and mask-free fusion. Our approach achieves cross-subject generalization differently. By leveraging graph spectral transforms, we project all subjects' fMRI into a shared frequency-informed latent space. This representation is more interpretable and potentially more transferable than implicit cycle-consistency or explicit mapping matrices, while remaining focused on brain-to-vision reconstruction. Frequency alignment is naturally preserved across subjects, and semantic guidance is injected via the brain-to-text projection layer.

**Multimodal brain-conditioned generation.** Beyond single-modality decoding, some approaches (Yang et al., 2024; Chang & Chen, 2021; Liu et al., 2024a; Ferrante et al., 2023) fuse brain activity with other inputs to enhance generation. MindCustomer (Yu et al., 2025a) synthesizes fMRI responses alongside external images or text for few-shot cross-subject adaptation. NeuroPictor (Huo et al., 2024) also incorporates shared semantic features to guide decoding. Our framework provides a complementary perspective. Rather than blending brain signals with external inputs, we emphasize the intrinsic frequency structure of fMRI and its direct alignment with image frequency bands. Text embeddings act as a semantic prior via the brain-to-text projection layer, interacting with frequency-masked fMRI to guide generation. This design improves reconstruction fidelity, preserves interpretability, and enables creative, semantically enriched outputs, reflecting the interpretive and imaginative aspects of human visual cognition.

## A.2  ADDITIONAL VISUALIZATIONS

Fig. 5 illustrates the effectiveness of our frequency-guided alignment strategy in reconstructing visual experiences from fMRI signals. The results highlight a complementary relationship between low-level and high-level reconstructions. Low-level outputs preserve coarse structural elements,

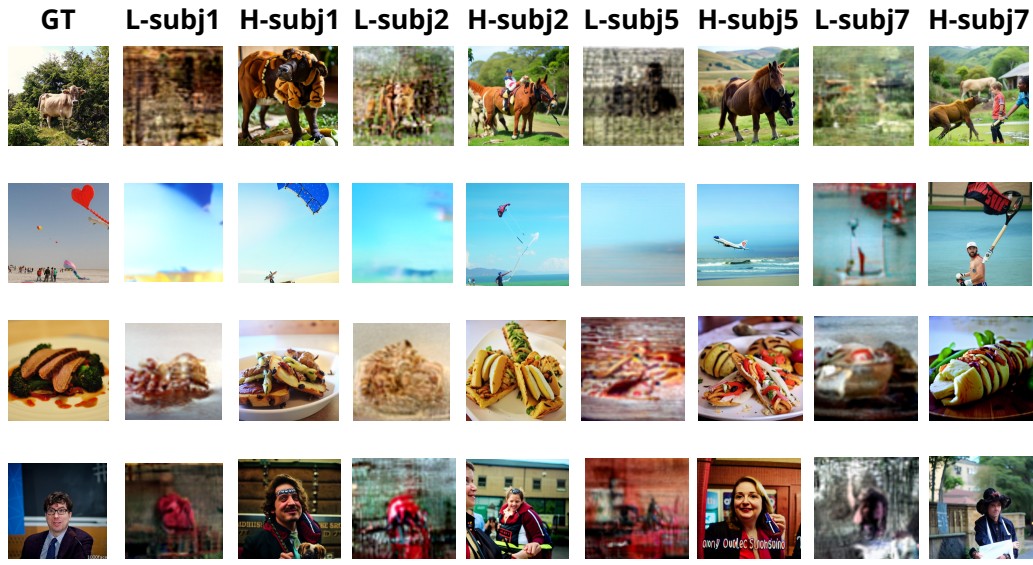

Figure 5: fMRI-to-image reconstruction with frequency-guided alignment. Low-level reconstructions capture coarse structures and layouts, while high-level reconstructions yield more realistic and semantically rich images.

such as object layout and spatial arrangement, providing a faithful representation of the global scene. In contrast, high-level reconstructions capture richer semantic content, producing more realistic and imaginative images that better align with human visual perception.

This dual-level reconstruction reveals two important insights. First, frequency-guided alignment enables a progressive refinement process: low-frequency components anchor the reconstruction with reliable structural cues, while high-frequency components enrich the result with semantic and contextual details. Second, the differences between low- and high-level reconstructions underscore the inherent challenge of decoding fMRI signals, where low-level alignment is more directly grounded in the neural signal, but high-level reconstruction benefits from the model's ability to leverage prior knowledge.

These results demonstrate that our method not only recovers structural information from neural data but also bridges toward semantically meaningful interpretations, offering a more complete understanding of how brain activity maps to perceived visual content.

## A.3 LLM USAGE DECLARATION

We disclose the use of Large Language Models (LLMs) as general-purpose assistive tools during the preparation of this manuscript. LLMs were used only for minor tasks such as grammar and style improvement, code verification, and formatting suggestions. No scientific ideas, analyses, experimental designs, or conclusions were generated by LLMs. All core research, methodology, experiments, and results were performed and fully verified by the authors.

The authors take full responsibility for all content presented in this paper, including text or code suggestions that were refined with the assistance of LLMs. No content generated by LLMs was treated as original scientific work, and all references and claims have been independently verified. LLMs did not contribute in a manner that would qualify them for authorship.

