# OpenReview forum: "When the Brain Sees Beyond Pixels: Creative Brain-to-Vision Reconstruction"
_ICLR.cc/2026/Conference — Submitted to ICLR 2026_

### Official Review · Reviewer_Kkq6 · 2025-10-27

**Soundness:** 2
**Presentation:** 2
**Contribution:** 3
**Rating:** 2
**Confidence:** 3

**Summary:**

This paper addresses limitations of traditional fMRI-to-image reconstruction (overemphasizing pixel fidelity over the brain’s encoding of abstraction, semantics, and imagination). It proposes a frequency-informed framework aligning neural, visual, and semantic modalities across bands: fMRI is decomposed via graph spectral transforms; images use masked frequency modeling; three lightweight frequency-aligned projection layers map fMRI to pretrained vision/language spaces (VDVAE, CLIP-Vision, CLIP-Text). Validated on NSD data, it achieves coarse-to-fine reconstruction with "creative alignment" (beyond pixel replication).

**Strengths:**

Unlike prior methods (e.g., Brain-Diffuser, MindBridge) that treat fMRI/image information uniformly, the framework explicitly leverages neuroscientific evidence (visual cortex’s frequency selectivity) to decompose fMRI into graph-spectral bands and images into Fourier bands. This alignment ensures biological plausibility and improves both structural (global layout) and semantic (object details) reconstruction fidelity (Table 1, Table 2).

**Weaknesses:**

1. Insufficient Visualization of Frequency Alignment Mechanism: Though claiming to "align fMRI graph-spectral components with visual frequency bands," the paper provides no direct evidence—no comparative visualizations of fMRI frequency components (e.g., low- vs. high-frequency graph modes) and their corresponding visual outputs (e.g., coarse layout vs. fine texture), nor dimensionality-reduced plots (e.g., t-SNE/UMAP) of frequency-masked fMRI/visual embeddings to show cross-modal clustering. This makes the "frequency alignment" mechanism a "black box" with limited interpretability.

2. Incomplete Quantitative Comparison with SOTA: It only compares with Brain-Diffuser (2023) and MindBridge (2024), missing comparisons with key recent methods like NeuroPictor (Huo et al., 2024) and MindEye2 (Scotti et al., 2024) on unified metrics (e.g., SSIM, CLIP, SWAV).

3. Inadequate Ablation of Core Modules: Existing ablations (Tables 1–2) only focus on frequency masking, lacking ablations of individual projection layers (e.g., removing the brain-to-text layer to test the necessity of semantic priors), validation of cross-attention in Versatile Diffusion (e.g., replacing it with concatenation), and comparisons of fusion strategies (e.g., attention weighting vs. current weighted average). This prevents distinguishing module contributions and proving the method’s optimality.

4. Limited Validation of "Creative Reconstruction": While emphasizing "creative reinterpretation," the paper lacks objective/subjective validation—no user study to assess if "creative" generations align with human perception of brain-encoded content, and no quantitative metric for "creativity" (e.g., generation diversity for the same fMRI input) to distinguish creativity from noise-induced distortion.

**Questions:**

NAN

---

### Official Review · Reviewer_37H2 · 2025-10-28

**Soundness:** 2
**Presentation:** 3
**Contribution:** 2
**Rating:** 4
**Confidence:** 4

**Summary:**

The paper's primary motivation is to shift the goal of fMRI-to-image reconstruction away from pure pixel-for-pixel replication of a perceived image. The authors argue that this traditional approach overlooks what brain signals fundamentally encode. The authors' goal is to create a new "frequency-informed framework" that bridges this gap by explicitly aligning the frequency structures of neural (fMRI), visual (image), and semantic (text) data.

**Strengths:**

The idea of this paper is interesting, which reframes the goal from pixel-perfect replication to a more meaningful "creative alignment" that decodes the brain's semantic interpretation of a scene. Methodologically, aligning the fMRI in the frequency domain is also interesting. The presentation of this paper is clear and good.

**Weaknesses:**

- The paper's main objective is to shift from pixel-perfect replication to "creative alignment" that captures abstraction and semantics. Even though it is interesting, it is very difficult to evaluate quantitatively. When the model produces an image that does not match the stimulus, it can be framed as a "creative reinterpretation" (a success) rather than an inaccurate reconstruction (a failure). The given evaluation metrics follow the previous work, which means that we are still evaluating the performance as in the previous work. And it is hard to justify the goal of the paper.

- From my understanding, the paper's central hypothesis is that fMRI "graph frequencies" (derived from the brain connectome) directly map onto image "2D Fourier frequencies" (derived from visual content), which imposes this structure:

   - Low-frequency fMRI → Low-level structure (VDVAE)

   - High-frequency fMRI → High-level semantics (CLIP)
Is there any neuroscientific evidence provided to support this direct 1:1 mapping? It's equally plausible that semantic information (e.g., "this is a cat") is encoded across multiple brain graph frequencies, not just "high" ones.

- About the results presented in Table 1. The core narrative is that low-frequencies handle structure and high-frequencies handle semantics. However, the quantitative results in Table 1 show that the "High-freq" only condition outperforms all others on both low-level metrics (like SSIM) and high-level metrics (like CLIP Score). Also in the text, the authors mentioned that "High-frequency inputs yield the best low-level and semantic fidelity, capturing fine details and object-specific attributes". What is the definition of the low-level features here? The conclusion "Our frequency-informed method preserves global layout and fine semantic details" is also difficult to observe in Figure 3.

- In the abstract, authors mentioned about capturing "imagination", which may be over-claiming, as the NSD data does not contain imagination data.

**Questions:**

My questions are included in the weaknesses above. I don't have other specific questions.

---

### Official Review · Reviewer_fSeW · 2025-10-31

**Soundness:** 2
**Presentation:** 3
**Contribution:** 2
**Rating:** 4
**Confidence:** 5

**Summary:**

The paper proposes a frequency-informed approach for generating images from fMRI signals. The method applies graph spectral transforms on fMRI signals and masked frequency modeling on images, and aligns low-, mid-, and high-frequency structures across the two data types. To ensure that the generation is meaningful, the authors use semantic priors using CLIP-text embeddings and multi-level visual features. Attention mechanisms are employed to allow the frequency-masked brain signals to interact with both the visual reconstructions and the textual cues. The low-level hierarchical brain-to-vision layer aligns masked fMRI signals with hierarchical probabilistic features extracted by an VDVAE encoder, capturing coarse structures and layouts. The high-level semantic brain-to-vision layer aligns masked fMRI signals with deterministic semantic features from the CLIP-Vision encoder. Finally, the brain-to-text alignment layer connects masked fMRI signals to CLIP-Text embeddings.

**Strengths:**

1. The masked frequency modeling for images and alignment between brain graph modes and visual spatial frequencies is a neat idea.
2. Ablation studies showing the correspondence between graph spectra and image frequency bands have been provided.

**Weaknesses:**

1. Complete Absence of Quantitative Comparisons to Prior Work
2. Questionable "Creative Alignment" Framing: The abstract and conclusion repeatedly claim the method enables "creative reinterpretation" and goes "beyond pixel replication" but they only evaluate on qualitative metrics and show handpicked examples to support their argument of "creative reinterpretation"
3. Contradiction with evaluation: Table 1 caption says "frequency-specific masking affects both structural fidelity and semantic alignment". But if you're aiming for "creative reinterpretation," why does fidelity matter?
4. Graph Spectral Transform is Underspecified: "edges E encode local anatomical or functional relationships." Which one is it? Anatomical or functional? If anatomical is it Euclidean distance in MNI space? Surface mesh? and if functional is it correlation of BOLD signals or functional clustering?

Minor:
Fig 2: the third box should be labeled as (c).

**Questions:**

1. Weaknesses as above.

2. In Step 6 of the algorithm, would a weighted loss be better?

---

### Official Review · Reviewer_SJ5P · 2025-11-01

**Soundness:** 3
**Presentation:** 2
**Contribution:** 3
**Rating:** 2
**Confidence:** 4

**Summary:**

This paper proposes a novel fMRI-to-image reconstruction framework that shifts the focus from pixel-level fidelity to the *creative alignment* of neural and visual representations. The central idea is to decompose both fMRI signals and images into frequency bands and explicitly align their components. Using graph spectral transforms on fMRI data and masked frequency modeling on images, the method aims to integrate coarse-scale structural information with higher-level imaginative features of human vision. Conceptually, this is an appealing direction—the notion of modeling the brain’s interpretive, imaginative processes is highly attractive. However, based on the presented results, it is difficult to distinguish between genuinely “creative” generations and failed reconstructions. While the multi-scale alignment inspired by neuroscience priors is imaginative, the current implementation does not convincingly achieve either high-quality reconstruction or the preservation of “imaginative” content. Consequently, the ambitious conceptual framing risks overstating the effectiveness of the proposed strategy, though this limitation may partly stem from dataset constraints and the inherent complexity of mapping brain representations.**

**Strengths:**

The primary strength of this paper lies in its conceptual contribution: introducing the novel and challenging problem of “creative alignment” in the fMRI-to-image reconstruction task. The core method leverages graph spectral transforms on fMRI signals and masked frequency modeling on images to align their respective frequency components. This alignment is implemented efficiently by training only three lightweight projection layers that map fMRI data into the latent spaces of frozen pretrained models. This approach enables a coarse-to-fine generation process that, on some level, is consistent with biological principles of hierarchical visual processing.

**Weaknesses:**

1. The central premise of "creative alignment" and "imaginative" reconstruction is conceptually appealing but operationally undefined. The paper does not propose a method for measuring this creativity. Instead, it relies on the same standard fidelity metrics (PixCorr, SSIM, AlexNet, CLIP) that it claims to be moving beyond. This creates a disconnect between the paper's motivation and its evaluation. If the goal is not pixel replication, why are pixel correlation and SSIM used as key metrics? The claims of "creativity" are currently subjective and unsubstantiated by the quantitative results.
2. While the qualitative comparisons to baselines (MindAligner, Brain-Diffuser, MindEye2, MindBridge) are excellent, the paper provides no quantitative comparison against these same methods. The quantitative tables (Tables 1 & 2) are only ablation studies of the authors' own model. Without a main results table showing how "Ours" performs against the baselines on standard metrics (PixCorr, SSIM, CLIP Score, etc.), it is impossible to verify the robustness of the qualitative claims.
3. The experiments use only the Natural Scenes Dataset (4 subjects) and do not test generalization beyond this controlled setting. It is unclear how the method would perform on other fMRI datasets or under more realistic conditions. No cross-subject or cross-dataset validation is provided. The reliance on one dataset may limit confidence in broad applicability.
4. The notion that reconstructions are “creative” or reflective of imagination is intriguing but subjective. The paper provides qualitative examples and observations, but no objective measure of creativity. In practice, a decoder could hallucinate details that are not in the neural data. It is unclear how one can disentangle true brain-driven inference from generative bias of the diffusion model.
5. Using CLIP and text priors helps avoid generic outputs, but it may also inject external biases. If the CLIP-text layer strongly influences the generation, some “reconstruction” could reflect common image-text associations rather than actual neural evidence. The balance between respecting the fMRI signal and imposing semantic priors is not fully explored.

**Questions:**

See the weakness section above.

---

### Meta-Review · Area_Chair_AcZ4 · 2025-12-19

**Summary:**

The decision to reject is based on a consensus among reviewers regarding critical flaws that undermine the paper's contributions. The central premise of "creative alignment" is methodologically ill-defined. Reviewers SJ5P, 37H2, and Kkq6 noted that without a rigorous definition or objective metric for "creativity," it is impossible to scientifically distinguish between a successful "imaginative" reconstruction and a simple model hallucination or failure. Also the submission does not provide quantitative comparisons against state-of-the-art baselines (e.g., MindEye2, NeuroPictor, Brain-Diffuser). As noted by Reviewers fSeW and Kkq6, the paper relies exclusively on internal ablations, making it impossible to benchmark its actual performance relative to the field.

**Reviewer Concerns:**

Authors did not provide a rebuttal.

**Reviewer Scores:**

Authors did not provide a rebuttal.

---

### Decision · Program_Chairs · 2026-01-26

Reject